# The impact of COVID-19 pandemic on ridesourcing services differed between small towns and large cities

**Nael Alsaleh**[ID]*, **Bilal Farooq**[ID]

Laboratory of Innovations in Transportation (LiTrans), Department of Civil Engineering, Toronto Metropolitan University, Toronto, Ontario, Canada

* nael.alsaleh@ryerson.ca

## Abstract

To curb the spread of the ongoing 2019 novel coronavirus (COVID-19), authorities have adopted several non-pharmaceutical (NPIs) and pharmaceutical interventions, which significantly affected our daily activities and mobility patterns. However, it is still unclear how severity of NPIs, COVID-19-related variables, and vaccination rates have affected demand for ridesourcing services, and whether these effects vary across small towns and large cities. We analyzed over 220 million ride requests in the City of Chicago (population: 2.7 million), Illinois, and 52 thousand in the Town of Innisfil (population: 37 thousand), Ontario, to investigate the impact of the COVID-19 pandemic on the ridesourcing demand in the two locations. Overall, the pandemic resulted in fewer trips in areas with higher proportions of seniors and more trips to parks and green spaces. Ridesourcing demand was adversely affected by the stringency index and COVID-19-related variables, and positively affected by vaccination rates. However, compared to Innisfil, ridesourcing services in Chicago experienced higher reductions in demand, were more affected by the number of hospitalizations and deaths, were less impacted by vaccination rates, and had lower recovery rates.

## Introduction

Since 2011, smartphone App-based ridesourcing services have emerged as a new travel mode in many cities worldwide [1–5]. Over the past decade since its inception, the service has seen a dramatic increase in popularity and market share due to features like the flexibility in the pickup and dropoff locations and times, safety, wide coverage areas, and reliability [5–8]. However, in March 2020, the 2019 novel coronavirus (COVID-19) began to spread rapidly across the globe, prompting the World Health Organization (WHO) to declare it a global pandemic [9–11]. Consequently, governments have introduced various Non-Pharmaceutical Interventions (NPIs) to curb the spread of the virus including lockdowns, stay-at-home orders, work-from-home policies, closing schools, universities, and non-essential workplaces, social distancing, wearing masks, and movement restrictions [12–15]. Additionally, by the beginning of 2021, different COVID-19 vaccines have been approved for emergency use by the WHO

**Data Availability Statement:** The minimal dataset underlying the results described in our paper, including all the data needed to reproduce the figures, values behind the descriptive statistics, and code to generate the demand models, is available

on a public repository at: https://github.com/
LiTrans/COVID-19-Impacts-on-Ridesourcing-
Services.git Researchers can access the full
operational data for ridesourcing services in
Chicago directly from the Chicago data portal at:
https://data.cityofchicago.org/Transportation/
Transportation-Network-Providers-Trips/m6dm-
c72p. The land-use and the detailed operational
data for ridesourcing services in Innisfil, however,
are third-party data that we cannot upload directly.
To access these datasets, researchers can contact
Paul Pentikainen, Senior Policy Planner-Town of
Innisfil, at (ppentikainen@innisfil.ca), and Matthew
Di Taranto, Senior Account Executive-Transit
Partnerships, at (416-628-3234) or at (matthew.
ditaranto@uber.com).

**Funding:** This research was funded by the Town of
Innisfil under grant number (1-51-48333) and
Canada Research Program in Disruptive
Transportation Technologies and Services (CRC-
2017-00038).

**Competing interests:** The authors have declared
that no competing interests exist.

and governments have started to immunize their population to reduce the risk of getting and transmitting the virus. Nevertheless, after more than two years since the first case was reported, COVID-19 is still an ongoing pandemic mainly due to the vaccine-hesitancy and inequality as well as the emergence of new variants [16, 17]. This raises an important question, how did the COVID-19 pandemic impact the transportation demand in general and the demand for ridesourcing services in particular?

Previous literature revealed that the pandemic has significantly influenced demand for all modes of transportation [18, 19] as well as changed individual's travel behaviour and preferences [20–23]. During the COVID-19 first and second waves, public transit, taxi, and ridesourcing services have experienced substantial reductions in the ridership [19, 24, 25]. However, public transit users were more likely to change their travel mode compared with the users of other modes, due to their concerns regarding the safety of the service and fears of getting infected [19, 20, 26–28]. Moreover, user's low confidence in public transit is expected to slow down the recovery of the system [29]. On the contrary, the use of private vehicles, motorcycles, and active means of transportation have become more common [19, 26].

The reduction of transportation demand during the early stages of the pandemic was mainly due to the COVID-19 related variables, e.g., new hospitalized cases, NPIs, as well as individual's attitudes and perceptions towards the pandemic. A recent study in China explored the impact of the pandemic on the behaviour of ridesourcing drivers using the actual data from September 2019 to August 2020. The study indicated that the daily number of new COVID-19 cases significantly affected the number of trips drivers made at the beginning of the pandemic. However, the effect of newly recorded cases decreased during the reopening phase [24]. Another recent study in the same country showed that the demand for both ridesourcing and taxi services was significantly affected by the number of COVID-19 cases, COVID-19 related policy measures, mean transportation cost, and the operational status of mass transit [19]. In the United States, social distancing measures and stay-at-home orders implemented in March 2020 resulted in nearly 30% reduction of personal trips [14]. The impact of the COVID-19 related interventions at the beginning of the pandemic was not limited to the motorized modes, but they have also significantly reduced the daily walking behaviour of people in many places. However, in the subsequent months, when the weather became warmer and some commercial activities resumed, recreational walking behaviour has increased markedly [30].

On the other hand, it was documented that the risk attitude of individuals was more influential on mobility than the actual mortality and hospitalization rates of COVID-19, especially during the early stages of the pandemic [31]. Using data obtained from a web-based survey in Greater Toronto Area (GTA) in July 2020, researchers showed that the perception of risk and safety has affected individual's decision of using ridesourcing services [25]. In another recent study, the authors examined the travel behaviour changes at the beginning of the pandemic in Java Island, Indonesia, using a web-based survey data. The results revealed that individuals who took preventative measures for the COVID-19 when leaving home were less likely to engage in outdoor activities and more likely to reduce the frequency of using ridesourcing services [32].

To summarize, various NPIs have been adopted to reduce individual's mobility which, in turn, played a major role in slowing the transmission of the virus [33, 34]. In a recent study, researchers used mobility data along with web search queries to identify the potential locations of COVID-19 outbreak hotspots [35]. Other studies simulated the impact of individual's mobility and vaccination rates on the evolution of the pandemic [36] or provided policy options for the recovery of public transit and shared mobility [37, 38]. Yet, it is still unclear whether the effects of the pandemic on ridesourcing services differed between small towns and large cities. Moreover, the implemented NPIs and their strictness varied across countries and

over time based on the epidemiological situation, the availability of the vaccines, vaccination rates, the existence of new variants, and the economic situation. Therefore, there is a strong need to examine the effects of the severity of NPIs and vaccination rates on the reduction in ridesourcing demand.

In the current study, COVID-19 related variables and NPIs, land-use data, socioeconomic and demographic characteristics, weather data, and the actual ridesourcing data for the City of Chicago, Illinois, and the Town of Innisfil, Ontario, from November 2018 to August 2021 are used to analyze and compare (a) the impacts of the COVID-19 pandemic on the spatio-temporal patterns of the demand, (b) the effects of the severity of the NPIs, COVID-19 related variables, and vaccination rates on the percent reduction in the daily demand, as well as (c) the main factors affecting the direct demand (origin-destination-pair) in the post-pandemic era (the period following the declaration of COVID-19 as a global pandemic). This work can be of use to policymakers as well as service providers to understand the future impact of the pandemic on ridesourcing services in terms of the recovery of the service, preparedness for additional waves of COVID-19 as well as other pandemics.

## Materials and methods

### Data collection

This study was motivated by the need to understand and compare COVID-19 pandemic impacts across urban areas within the same geographical region. The Town of Innisfil and the City of Chicago were chosen as case studies. Several factors governed the selection of these case studies, including a) the availability of ridesourcing data, b) the coverage of both the pre-pandemic and pandemic periods, and c) the location of both case studies within the same region (North America). The Town of Innisfil is located on the western shore of Lake Simcoe in Simcoe County, Ontario, with a population density of 139 people per square kilometer. According to the 2016 census data, the town has a population of 36,566 people, around 67% of whom are in the working age group (15 to 64 years old). The median household income (after-tax) in Innisfil is $57,846, which is higher than the national average of $48,895 [39]. On the other hand, the City of Chicago is located on the southwestern tip of Lake Michigan in Illinois and is considered among the largest cities in the United States, with a population density of 4,618 persons per square kilometer. The City of Chicago has a population of more than 2.7 million and almost 70% of the population belongs to the working age group. The median household income (after-taxes) in Chicago is $62,613, which is lower than the national average of $70,690 [40].

In May 2017, the Town of Innisfil partnered with Uber Technologies Inc., the only transportation network provider (TNP) in the town, to provide residents with subsidized ridesourcing trips rather than operate a fixed-route transit system [41, 42]. Therefore, the service has been used for a wide variety of trip purposes including work, school, shopping, social, recreation, and medical appointments [43]. On the other hand, ridesourcing services are one of the main modes of travel in Chicago that operate independently of the public transportation service and are used mainly for non-work purposes [44]. We hypothesize that the differences in use purposes, operating policies, land area, and population density resulted in the COVID-19 pandemic having different effects on ridesourcing services between the two locations.

In this study, data from multiple sources were utilized to examine this hypothesis. Weather data were collected from AccuWeather Inc. [45] for both case studies. The collected dataset contained information on the daily average temperature, precipitation, and snowfall for the period of March 1, 2020 to July 31, 2021. Socioeconomic and demographic factors were obtained from the 2016 Statistics Canada and the 2017 American Community Survey for Innisfil and Chicago, respectively. Several characteristics were collected for both case studies at

the census tract level (CT), including population density, median income, gender (percentage of males), percentage of elderly people, education, marital status, and mode of commuting. Land-use data were obtained from the Chicago Metropolitan Agency for Planning and the Town of Innisfil. The COVID-19 related variables were acquired from the Simcoe Muskoka District Health Unit and Chicago COVID Dashboard for the period of January 1, 2020 to July 31, 2021. The main COVID-19 related variables considered in this study were the daily confirmed cases, deaths, hospitalizations, test positivity rate (percentage of tests performed that were positive for COVID-19), as well as the cumulative percentage of residents who received the first and second vaccine doses. As for the COVID-19 NPIs, several interventions have been implemented in Ontario and Illinois since the beginning of the pandemic, including the declaration of a state of emergency, closing schools, restaurants, and bars. Since the beginning of the pandemic, Ontario has declared the state of emergency (stay at home order) three times. The first state of emergency was declared on March 17, 2020, and ended on July 24, 2020. On January 14, 2021, a second state of emergency was announced, and it ended on February 19, 2021. The last state of emergency was declared on April 8, 2021, and ended on June 9, 2021. These states of emergency were preceded by the closure of schools, restaurants, bars, and non-essential workplaces [46]. On the other hand, Illinois declared only one state of emergency on March 21, 2020, which lasted until May 29, 2020. The state of emergency was followed by the reopening of schools and relaxation of most COVID-19-related measures by the end of June. Then several cycles of restaurant and bar closures and reopenings took place between the period of August 2020 and May 2021. In May 2021, fully vaccinated people were exempted from wearing mask for most indoor activities [47]. We used used Stringency Index data from the Oxford COVID-19 Government Response Tracker (OxCGRT) [48] to represent the severity of the government's policy measures in both locations. The data was collected from January 1, 2020, to July 31, 2021, for both case studies.

The actual ridesourcing trip data for Chicago were acquired from Chicago Data Portal [49] form November 1, 2018 to July 31, 2021, including 220,527,209 observations (ride requests). Each observation contained, amongst others, information about trip request date, pickup and dropoff times and CT, as well as trip distance, duration, and fare. On the other hand, the actual ridesourcing data for the second case study was provided through Uber Technologies, Inc. and the Town of Innisfil. The data obtained from Uber were for the pre-pandemic period and included information about the average hourly demand for a typical weekday and weekend as well as the average daily demand per month from May 2017 till February 2020. However, for consistency, only the data from November 2018 till February 2020 were used in this study. Moreover, the average hourly demand data were provided for five time periods: morning (6:00 AM to 10:00 AM), midday (10:00 AM to 3:00 PM), late afternoon (3:00 PM to 7:00 PM), evening (7:00 PM to 10:00 PM), and night (10:00 PM to 6:00 AM).

The Town of Innisfil provided us with the operational data for Uber service over the period of September 1, 2020 to July 31, 2021, including 52,126 trip requests. The operational dataset contained detailed information on trip request time, pickup and dropoff times and location, as well as trip distance, duration, and fare. Note that the dataset does not cover the first wave of the COVID-19 outbreak; however, it can be useful to capture the effects of both the second and third waves, which had higher impacts on the healthcare system, as well as the vaccination rates on the ridesourcing service in Innisfil.

## Methods

**Stringency index.** Oxford COVID-19 Government Response Tracker (OxCGRT) gathers publicly available data on 23 indicators of policy measures governments have taken to respond

to the pandemic from more than 180 countries. There are eight indicators related to the government's containment and closure policies (C1-C8), eight indicators related to the health system policies (H1-H8), four indicators related to economic policies (E1-E4), and four indicators related to vaccine policies (V1-V4). The data from the 23 indicators are used to develop five main indices: overall government response index, containment and health index, stringency index, economic support index, and risk of openness index. The stringency index represents the strictness of a government's policy measures taken over time to restrict individual's movements and behaviour. It is calculated based on nine indicators related to containment, closure, and health policies, including: (a) school and university closure, (b) workplace closure, (c) public event cancellations and restrictions, (d) restrictions on private gatherings, (e) public transport closures, (f) stay-at-home orders, (g) restrictions on intra-provincial travel, (h) restrictions on international travel, and (i) public information campaigns. The stringency index is simply the arithmetic mean of the nine indicators. It ranges from 0 to 100, with 100 representing the most stringent response [48, 50].

$$stringency\ index = \frac{1}{N}\sum_{i=1}^{N} Indicator_i \tag{1}$$

where $N$ is the number of indicators used in the index, and $Indicator_i$ is the sub-index score for each indicator. Each sub-index score (Indicator) for any given indicator (i) on any given day (t), is calculated as follows:

$$Indicator_{i,t} = 100\frac{Value_{i,t} - 0.5(Flag_i - BinaryFlag_{j,t})}{MaxValue_i} \tag{2}$$

where $Value_{i,t}$ is the recorded policy value on the ordinal scale. $Flag_i$ is a flag variable corresponds to the geographic scope of the policy (1 if the policy is general and 0 if targeted). $BinaryFlag_{j,t}$ is the recorded binary flag for the indicator, if the indicator has a flag. $MaxValue_i$ is the maximum value of the indicator [48, 50].

**Random forest regression algorithm.** Random Forest algorithm is a non-linear statistical ensemble method widely used for both regression and classification. For regression tasks, multiple de-correlated decision trees ($N$) are constructed at training time, and the mean of the predictions from the trees ($\hat{Y}(x_i)$) is calculated to produce more consistent and accurate outcomes than using a single tree ($DT_n(x_i)$). To avoid the possibility of over-fitting, a bootstrap sample of the training data (input space) is selected with replacement for every single tree, and a random subset of the available features is selected for splitting (feature space) at each node. The node is then split based on the feature that minimizes the cost function, such as the sum of squared errors. This process continues iteratively until either the predefined maximum depth is reached, or a terminal node is reached. Final predictions are calculated by averaging the results from the individual trees [51, 52].

$$\hat{Y}(x_i) = \frac{1}{N}\sum_{n=1}^{N} DT_n(x_i) \tag{3}$$

There are several performance metrics to evaluate the predictive accuracy of regression models, but the following four are commonly used: Mean Absolute Error (MAE), Mean Squared

Error (MSE), Root Mean Squared Error (RMSE), and Coefficient of Determination ($R^2$).

$$MAE = \frac{1}{K}\sum_{k=1}^{K}|\hat{Y}_k - Y_k| \qquad (4)$$

$$MSE = \frac{1}{K}\sum_{k=1}^{K}(\hat{Y}_k - Y_k)^2 \qquad (5)$$

$$RMSE = \sqrt{\frac{1}{K}\sum_{k=1}^{K}(\hat{Y}_k - Y_k)^2} \qquad (6)$$

$$R^2 = 1 - \frac{\sum_{k=1}^{K}(\hat{Y}_k - Y_k)^2}{\sum_{k=1}^{K}(\bar{Y} - Y_k)^2} \qquad (7)$$

where K is the number of observations of the training sample, $Y_k$ represents the nth observed value of the dependent variable, $\hat{Y}_k$ is the nth predicted value of the dependent variable, and $\bar{Y}$ is the average value of the dependent variable.

**Bayesian optimization approach.** It is necessary to fine-tune the hyperparameters of a particular machine learning model to find its optimal architecture and, therefore, performance. Traditionally, tuning hyperparameters involves setting and testing several combinations of hyperparameters manually. This method, however, becomes inefficient and time-consuming when many hyperparameters are being evaluated. Alternatively, automatic tuning approaches can be used, like grid search, random search, and Bayesian optimization. The Bayesian optimization approach involves manually creating a hyperparameter space from which the algorithm can select a set of hyperparameters for model training. A cross-validation score is then returned by the trained model, and the surrogate function uses the score to recommend new hyperparameters to the objective function that are likely to enhance the performance. This iterative process is repeated until the predefined number of iterations is achieved. Compared with other automatic tuning approaches, this approach typically takes fewer evaluations to find the optimal hyperparameter set [53].

**SHAP analysis technique.** SHapley Additive exPlanations (SHAP) [54] is a game theoretic approach used to explain the predictions of machine learning models. Specifically, it explains the effect of each feature on the model output and its relative importance based on Shapely values. The Shapely value is calculated by taking the difference between the average model predictions and the predictions of a respective sample. In other words, Shapley values for a specific feature are estimated by comparing the model's predictions with and without that feature. The sign of the Shapely value indicates whether the corresponding feature value has a positive or negative impact on the model's prediction, while magnitude represents how much contribution it makes to prediction. Therefore, the importance of a feature is computed by summing the absolute Shapley values of the feature across the testing dataset samples [55–57].

## Data analysis and modelling

Fig 1 demonstrates the methodological framework used in this study. As Fig 1 depicts, the current study involved three main parts. In the first part, we used the actual ridesourcing data to compare the impacts of the COVID-19 pandemic on the ridesourcing services in the Town of Innisfil and the City of Chicago in terms of the hourly, daily, and monthly distributions of the

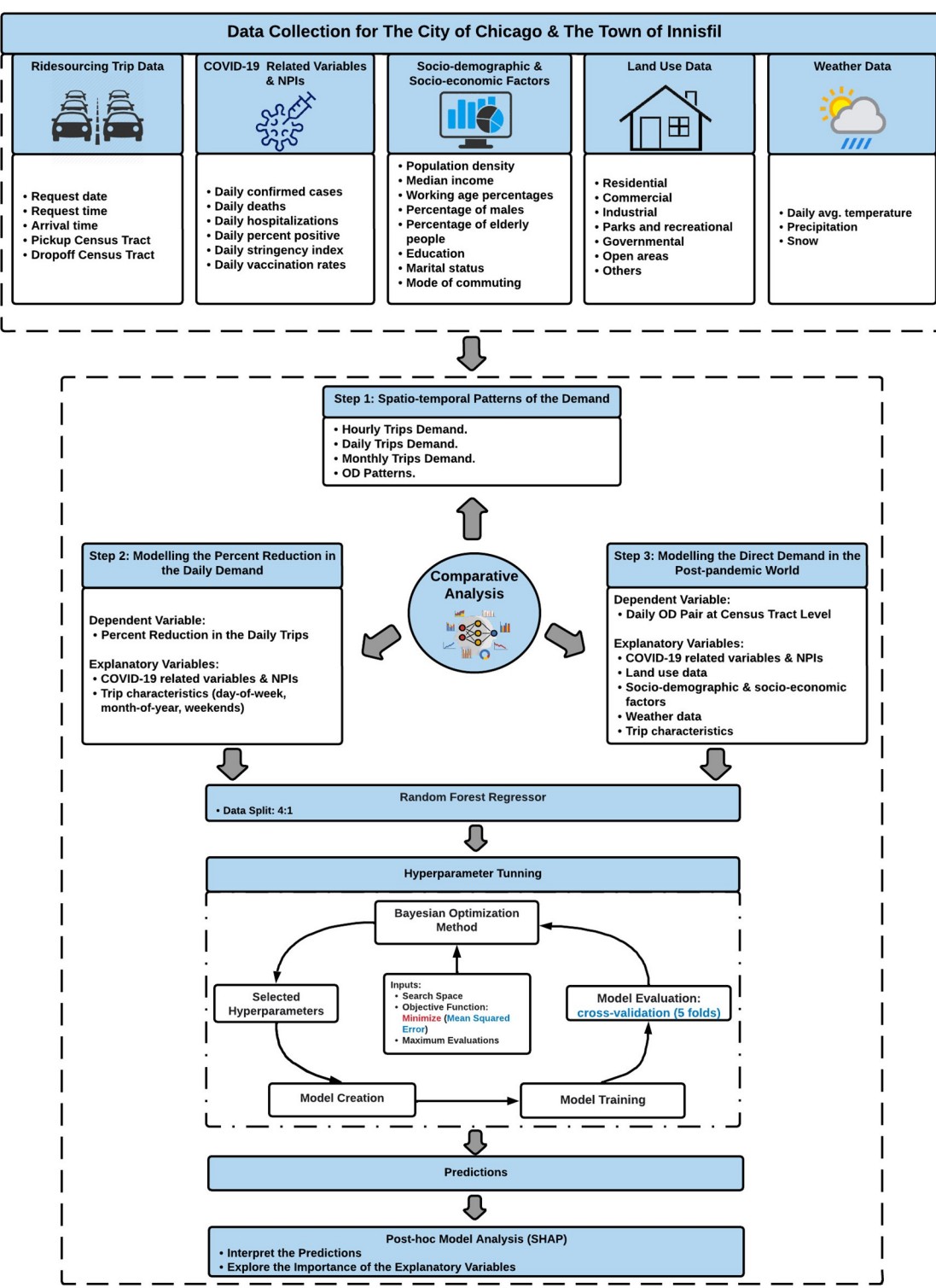

**Fig 1. Research framework.**

demand, as well as the origin-destination (OD) flow patterns. In the second part, we explored the impact of the COVID-19 stringency index on the daily and hourly demand values. Moreover, the actual ridesourcing data as well as the COVID-19 related variables and stringency index were used to model the percent reduction in the daily demand for the ridesourcing services in Innisfil and Chicago. In both models, the dependent variable represented the percent reduction in the daily demand values at the town/city level, which were computed based on the average weekday and weekend demand values for the pre-pandemic period (from November 1, 2018 to February 28, 2020).

$$Y_{DR_t} = 100 \frac{TD_t - A_{WE} \cdot WE_t - A_{WD} \cdot WD_t}{A_{WE} \cdot WE_t + A_{WD} \cdot WD_t} \tag{8}$$

where $Y_{DR_t}$ is the percent reduction in demand on any given day during the pandemic (t). $TD_t$ is the total number of ridesourcing trips in the town/city. $A_{WE}$ is the pre-pandemic average number of ridesourcing trips on weekends. $WE_t$ is a binary variable representing weekends (1 for weekends and 0 for weekdays). $A_{WD}$ is the pre-pandemic average number of ridesourcing trips on weekdays. $WD_t$ is a binary variable representing weekdays (1 for weekdays and 0 for weekends).

On the other hand, the explanatory variables used in each model included the COVID-19 related variables and stringency index as well as the key trip characteristics, such as: the day-of-week, month-of-year, and weekend dummy variable (to distinguish between weekdays and weekends). In the third part, we modelled the daily direct demand for ridesourcing services in each case study during the pandemic at CT level using COVID-19 related variables and stringency index, land-use data, socioeconomic and demographic characteristics, weather data, and the actual ridesourcing data. In both models, the dependent variable was formed by aggregating trips data, from March 1, 2020 to July 31, 2021 for the City of Chicago and from September 1, 2020 to July 31, 2021 for the Town of Innisfil, at the OD-pair level on each day. Table 1 provides descriptive statistics for both the dependent and explanatory variables.

$$Y_{ij_t} = \sum T_{ij_t} \tag{9}$$

where $Y_{ij_t}$ is the direct demand between two CTs on any given day during the pandemic (t). $T_{ij_t}$ is the total number of ridesourcing trips from $CT_i$ to $CT_j$.

The percent reduction in the daily demand and the direct demand models were developed using the Random Forest (RF) regression algorithm. In all models, data were randomly divided into training and test sets with 4:1 ratio. Moreover, Bayesian optimization approach was used to find the optimal architecture of each model. The Bayesian optimization approach was implemented using the "fmin" function. We used the Tree Parzen Estimator (TPE) algorithm for the surrogate function, and 65 for the maximum number of iterations. In each iteration, trained models were evaluated using 5-fold cross-validation with the objective function of minimizing the MSE. The best hyperparameter set for each model is presented in S1 Table. The optimal architectures of each model were applied on the testing dataset, and the following performance metrics were computed: MAE, MSE, RMSE, and $R^2$ (presented in S2 Table). Finally, the SHAP analysis technique was used to interpret the predictions of each model.

## Results

### Spatio-temporal demand patterns

Fig 2 presents the hourly, daily, and monthly distributions of the demand for ridesourcing services in Innisfil and Chicago pre and during the COVID-19 pandemic. Overall, it is observed

**Table 1. Descriptive statistics for the explanatory and dependent variables.**

| Variable | Unit | Town of Innisfil | | | | City of Chicago | | | |
|---|---|---|---|---|---|---|---|---|---|
| | | Mean | Std. Dev. | Min | Max | Mean | Std. Dev. | Min | Max |
| Dependent Variables | | | | | | | | | |
| Reduction in daily demand | % | 28.09 | 18 | 0.01 | 83.61 | 79.29 | 15.01 | 4.58 | 98.90 |
| Daily OD-pair trips | Count | 4 | 4.32 | 1 | 40 | 2.39 | 5.34 | 1 | 196 |
| Explanatory Variables | | | | | | | | | |
| Sociodemographic & socioeconomic | | | | | | | | | |
| Median household income | $ | 60170.9 | 6987.94 | 48846.7 | 68190.72 | 60489.7 | 34451.63 | 11146 | 194167 |
| Population density | per $km^2$ | 433.3 | 534.94 | 29.6 | 1526.1 | 7183 | 10335.21 | 152.5 | 266431 |
| Percentage of males | % | 50 | 2.11 | 45.3 | 51.6 | 48.3 | 5.03 | 25.2 | 90.3 |
| Percentage of working age group | % | 66.69 | 9.35 | 45.9 | 72.7 | 69.4 | 8.2 | 44.1 | 94.6 |
| Percentage of elderly people | % | 17.7 | 13.03 | 9 | 46.7 | 12.7 | 6.61 | 0 | 53.1 |
| Percentage of married people | % | 51.7 | 4.17 | 46.5 | 58 | 35.2 | 13.35 | 0 | 68.6 |
| Percentage of highly educated people | % | 12.6 | 3.95 | 8.3 | 19 | 36.4 | 26.3 | 0.9 | 94.8 |
| Percentage of personal vehicle commuters | % | 10.3 | 1.94 | 8.3 | 13.8 | 45.1 | 14.5 | 10.7 | 79.6 |
| Average household size | per hhld | 2.7 | 0.33 | 2 | 3 | 2.6 | 0.58 | 1.3 | 4.4 |
| Land use variables | | | | | | | | | |
| Residential land use ratio | % | 17.5 | 13.58 | 1 | 40 | 18 | 9.58 | 1 | 52.1 |
| Commercial land use ratio | % | 0.02 | 0.05 | 0 | 0.12 | 3 | 2.79 | 0 | 29.5 |
| Governmental land use ratio | % | 0.01 | 0.04 | 0 | 0.1 | 3.2 | 5.37 | 0 | 42.6 |
| Industrial land use ratio | % | 0.21 | 0.43 | 0 | 1.2 | 2.1 | 4.74 | 0 | 37.8 |
| Parks land use ratio | % | 0.9 | 0.84 | 0 | 2.1 | 0.4 | 1.94 | 0 | 25.3 |
| Weather variables | | | | | | | | | |
| Average daily temperature | C | 6.8 | 9.90 | −16 | 25 | 13.2 | 10.43 | −17 | 30 |
| Precipitation | mm | 0.2 | 0.52 | 0 | 3.51 | 0.2 | 0.72 | 0 | 9.32 |
| Snowfall | cm | 1.4 | 6 | 0 | 40.6 | 0.3 | 2.22 | 0 | 40.4 |
| COVID-19 related variables & NPIs | | | | | | | | | |
| Cases | Count | 3.5 | 3.77 | 0 | 19 | 558 | 572.49 | 0 | 3360 |
| Deaths | Count | 0.1 | 0.25 | 0 | 1 | 10.9 | 11.80 | 0 | 58 |
| Hospitalization | Count | 0.5 | 0.69 | 0 | 3 | 55.7 | 46 | 1 | 203 |
| Positive rate | % | 2 | 1.28 | 0 | 5 | 8.2 | 7.43 | 0 | 33 |
| Percentage of partially vaccinated | % | 21.3 | 28.2 | 0 | 77.28 | 13.96 | 21.29 | 0 | 58.7 |
| Percentage of fully vaccinated | % | 7.5 | 15.08 | 0 | 64 | 10.7 | 18 | 0 | 52.3 |
| Stringency index | % | 68.8 | 15.4 | 41.7 | 90.7 | 56.8 | 14.49 | 5.6 | 82.4 |
| Trip related variables | | | | | | | | | |
| Day of the week | – | – | – | 1 | 7 | – | – | 1 | 7 |
| Weekend | – | – | – | 0 | 1 | – | – | 0 | 1 |
| Month of the year | – | – | – | 1 | 12 | – | – | 1 | 12 |
| year | – | – | – | 0 | 1 | – | – | 0 | 1 |

The dependent variables, weather variables, and COVID-19 related variables and NPIs statistics were calculated based on the period from September 1, 2020, to July 31, 2021, for the Town of Innisfil, and March 1, 2020, to July 31, 2021, for the City of Chicago. Percentage of highly educated people variable represents the percentage of population with at least an undergraduate degree. Stringency index values are for Ontario and Illinois. Day of the week is a discrete variable represents the day of the week from Monday to Sunday. Weekend is a binary variable to distinguish between weekends and weekdays, with 1 for weekends. Month of the year is a discrete variable represents the month of the year from January to December. Year is a binary variable to distinguish between 2020 and 2021, with 1 for 2021.

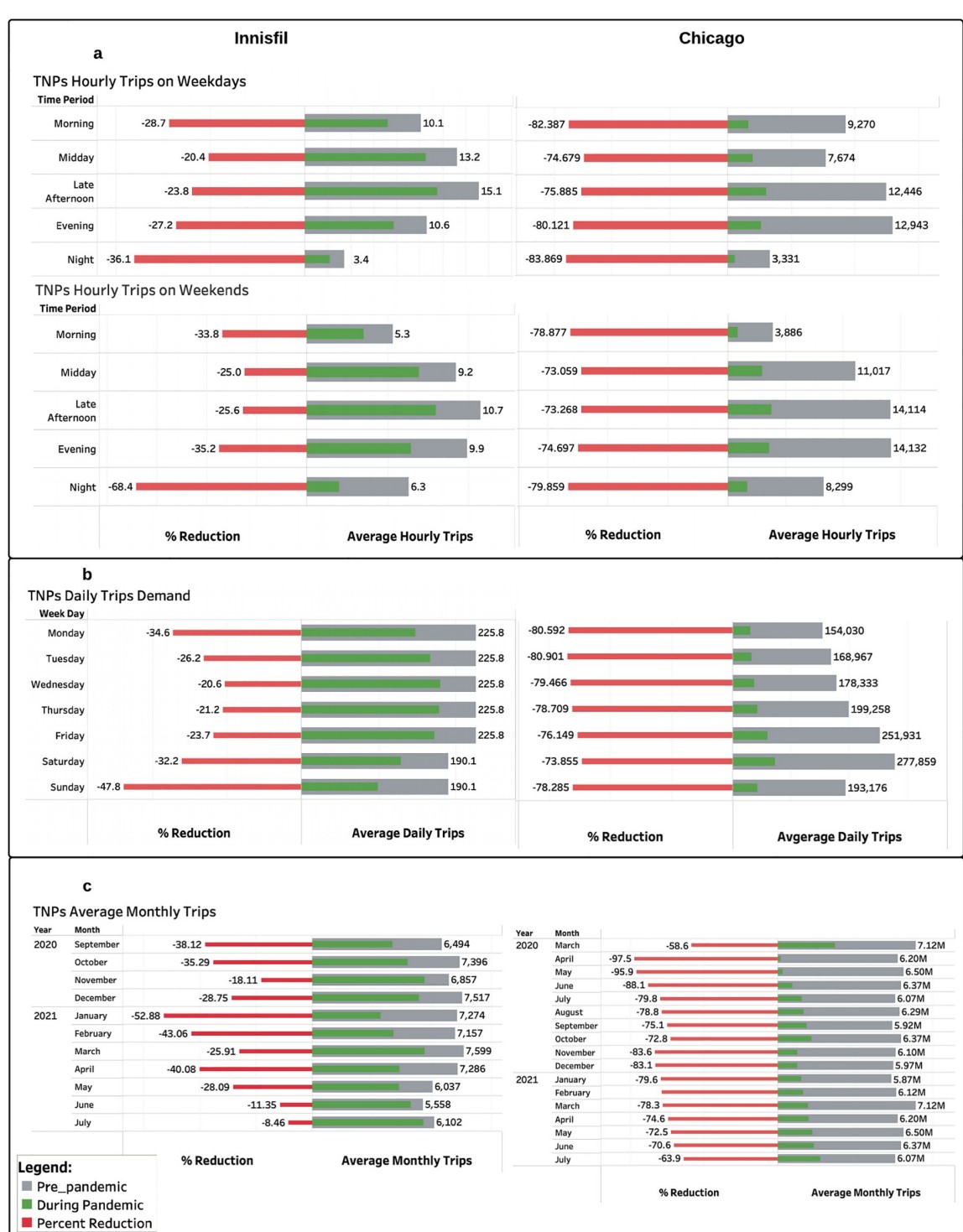

**Fig 2. COVID-19 pandemic impacts on the temporal distribution of the demand for the ridesourcing services in the Town of Innisfil and the City of Chicago.** (a) Hourly distribution of the demand pre and during the pandemic. (b) Daily distribution of demand pre and during the pandemic. (c) Monthly distribution of demand pre and during the pandemic.

that the demand in Chicago is much higher than in Innisfil. This is most likely due to the considerable difference in the population density and land area as well as in the attractions and recreational activities available in both locations. The average daily demand in Chicago was about 203,365 trips in the pre-pandemic period and 45,127 trips during the pandemic, while in Innisfil, the average values were about 216 and 153 trips per day for the pre and during pandemic periods, respectively. Before the pandemic, the demand was highest during the late afternoon period in Innisfil and during the late afternoon and evening periods in Chicago on both weekdays and weekends. Among the days of the week, the demand was the highest during the weekdays in Innisfil and during the weekends in Chicago. These patterns remained the same during the pandemic. However, the pandemic had a strong impact on the demand levels, and these effects varied significantly between the two locations.

On average, the hourly and daily demand values in Innisfil were reduced by 30% during the pandemic, whereas the hourly and daily demand values in Chicago were 80% less compared with the pre-pandemic levels. This might be due to the difference in the use purposes of the ridesourcing services between the two locations. The highest reduction in the demand in Innisfil on both weekdays and weekends was during the night period. Moreover, among the days of the week, weekends experienced higher reduction level in the demand. In Chicago, however, we observed a consistent and sharp decline in demand across all times of the day and across all days of the week. Even though the pandemic affected all trips, these findings indicate that non-work related trips were most impacted by the closure of non-essential businesses. It can also be noticed that the reduction levels in the monthly demand varied overtime. Both locations encountered a repeated pattern of a sharp decrease in the demand followed by a gradual recovery. In Innisfil, the highest reduction in the monthly demand was observed in January 2021 during the second wave of COVID-19 at 52.9% and the lowest in July 2021 following the end of the third wave at 8.5%. On the other hand, Chicago experienced the highest decrease in the monthly demand during the first wave in April 2020, at 97.5%, and the lowest in March 2020 at the beginning of the pandemic, at 58.6%.

Fig 3 displays the origin-destination flow patterns for the ridesourcing services in Innisfil and Chicago during the pandemic. In Innisfil, most of the OD pairs with high demand were found within the central area as well as between the central area and the neighbouring CTs containing the train stations. These patterns are not surprising, since the central area has the highest population density and contains several shopping malls, office buildings, and large grocery stores. On the other hand, the OD pairs associated with the train stations are likely to be the morning and afternoon commutes. In Chicago, the high-demand OD pairs were within the downtown area and between the downtown area and O'Hare International Airport. Chicago's downtown area is characterized by high population density, median income, and employment levels. The area is also home to large shopping malls, parks, hotels, as well as recreation and commercial areas. These factors have a positive effect on the ridesourcing demand [58]. O'Hare, one of the busiest and largest airports in the world, is expected to be one of the most popular destinations for ridesourcing. Finally, the pre-pandemic ridesourcing trips showed the same patterns in both locations (see S1 Fig).

## Factors influencing the percent reduction in the daily demand

Here, we investigate the effects of the COVID-19 related variables, government stringency index, and the key trip characteristics on the daily demand for the ridesourcing services. Fig 4a illustrates the relationship between the stringency index, vaccination rates, COVID-19 cases, and the daily trip demand in Innisfil and Chicago. In general, the increase in the stringency index values was associated with a decline in the daily trip demand as well as the COVID-19

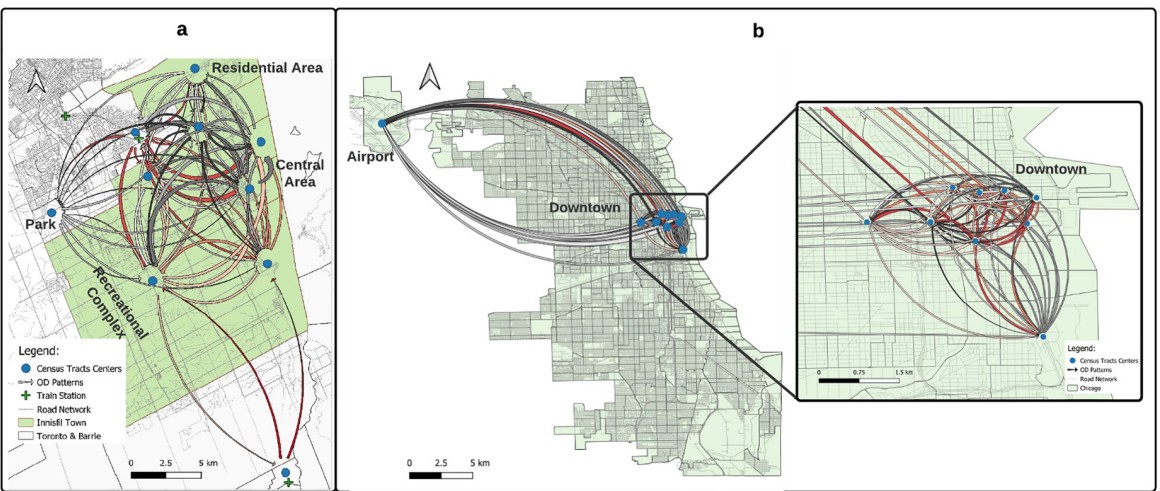

**Fig 3. Origin-destination (OD) flow patterns for ridesourcing services during the pandemic in (a) Innisfil and (b) Chicago.** In Innisfil, 52,126 ridesourcing trips were made from September 2020 till August 2021 covering 20 CTs, while more than 21 million trips were made in Chicago during the pandemic distributed over 803 CTs. Since it is hard to develop meaningful flow patterns for all OD pairs, we only display the flow patterns for the 10 most frequently used CTs in both locations. The OD flows shown in this figure represent 90% and 10% of the total trips in Innisfil and Chicago, respectively. The OD line color represents the origin of trips and the width reflects the flow strength. The wider the line is, the more trips the OD pair has. This figure was generated using the free and open-source software QGIS under the CC BY 4.0 license.

cases reported 2 weeks later. Ridesourcing services saw a significant decrease in demand immediately after each jump in the stringency index, followed by a gradual recovery. However, the extent of the stringency index effect varied among the two locations, and it had a noticeably higher impact on the ridesourcing demand in Chicago. Moreover, ridesourcing demand in Chicago showed a lower recovery rate, when compared to that in Innisfil. We argue that these trends are explainable by the differences in the use purposes and the operating policy between the two locations. High stringency index values have primarily restricted non-mandatory trips which represent a high proportion of the total trips served by the ridesourcing services in Chicago. Furthermore, it is noticed that the increase in the daily vaccination rates, first and second vaccine doses, was correlated with an increase in the ridesourcing trip demand and a decrease in the COVID-19 cases in both locations.

The impact of the stringency levels (less than 25, 25–50, 50–75, and more than 75) on the weekdays and weekends hourly demand values are shown in Fig 4b. We found that the hourly demand values continuously reduced with the increase in the stringency level. The extent of the stringency level effect was homogeneous across the day periods during weekdays and higher during the night period on weekends. Moreover, the reduction amount in the hourly demand was consistent with the transition from one stringency level to the next in Innisfil, whereas the second stringency level had the highest impact on the hourly demand values in Chicago.

Furthermore, the percent reduction in daily demand was modelled to get a better understanding of how the trip characteristics as well as the COVID-19 related variables and stringency index have affected the daily demand in Innisfil and Chicago. Fig 5 presents the SHAP summary plots for the percent reduction in the daily demand models. The results revealed that the stringency index, COVID-19 deaths, hospitalizations, and test positivity rate variables had a negative effect on the demand reduction levels in both locations. Daily demand values experienced higher reduction levels as the stringency index, COVID-19 deaths, hospitalizations, and test positivity rate increased. Regarding the effects of the trip characteristics variables,

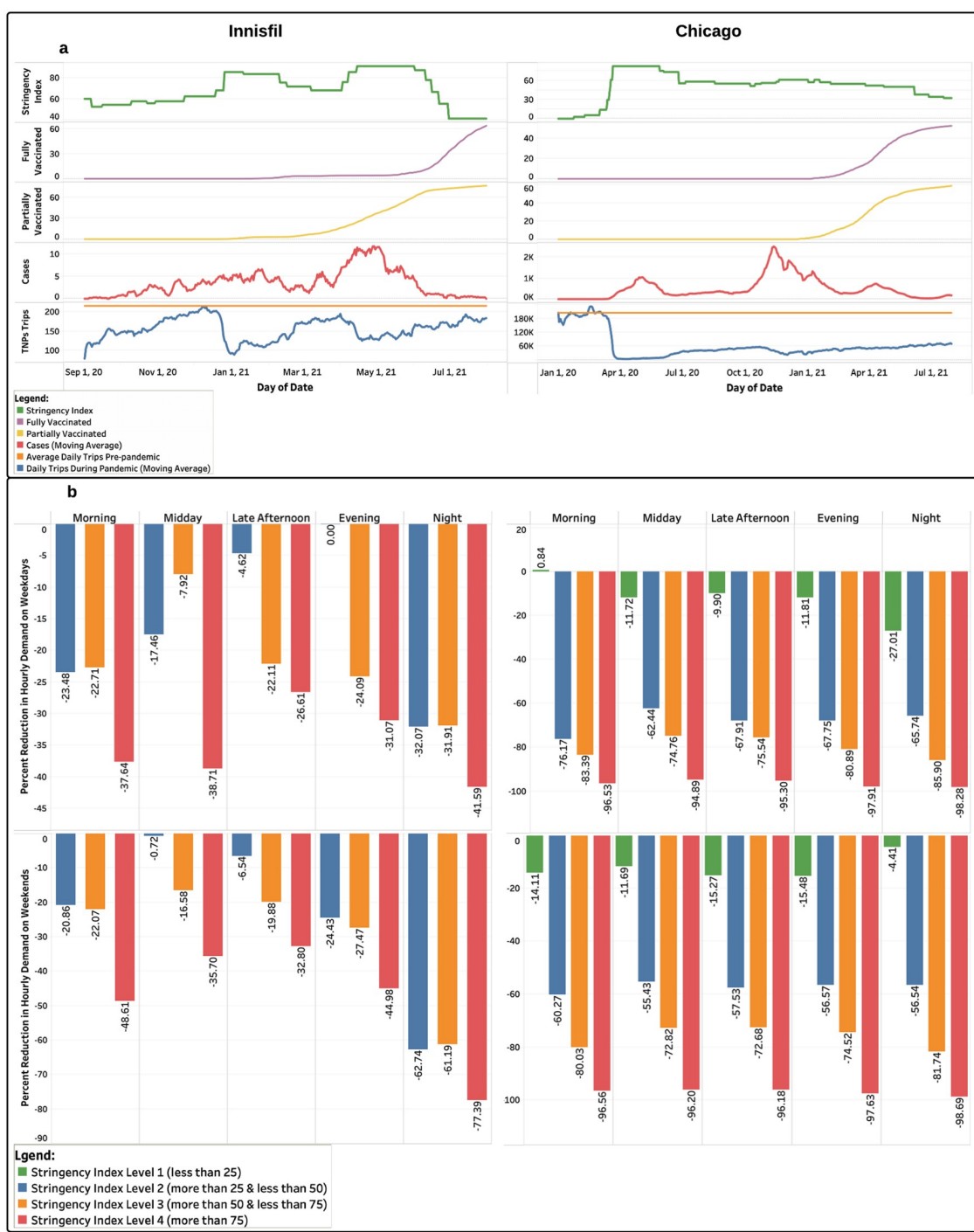

**Fig 4. The impact of stringency index on the daily and hourly demand values in Innisfil and Chicago.** (a) The relationship between the stringency index, vaccination rates, COVID-19 cases, and the daily trip demand in Innisfil and Chicago. (b) Effects of stringency index on weekdays and weekends hourly demand values in Innisfil and Chicago. To explore the impact of the stringency index on the hourly demand, we categorized its values into five levels: pre-pandemic (stringency index = 0), less than 25, 25–50, 50–75, and more than 75.

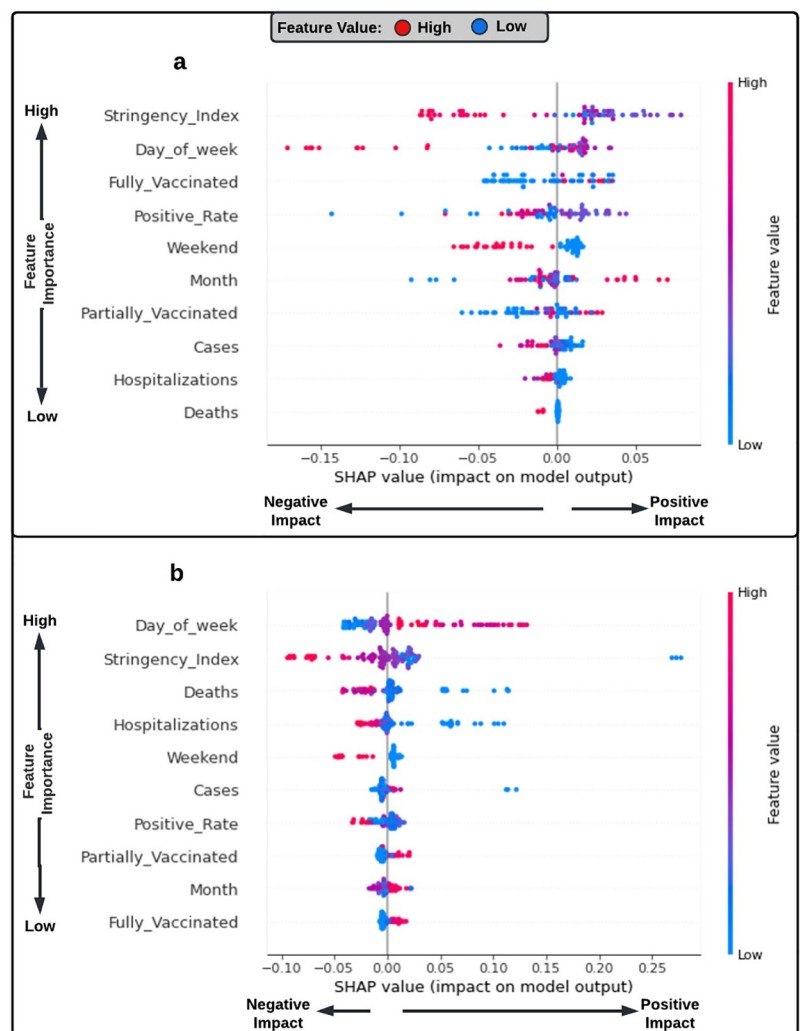

**Fig 5. SHAP summary plot for the percent reduction in the daily demand models.** (a) Innisfil model. (b) Chicago model. Each point in the summary plot represents a Shapley value for a feature and an instance. The feature importance determines its position on the y-axis, the Shapley value determines its position on the x-axis, and the value determines its colour.

weekend variable was found to have a negative impact, while both the day-of-week and month-of-year variables had a positive effect on the ridesourcing demand. These findings indicate that the ridesourcing demand saw higher reduction levels on weekends as well as at the beginning of the week and the year. Vaccination rate variables, on the other hand, had a positive impact on the daily demand values. Daily demand values increased with the increase in the number of partially and fully vaccinated individuals. This suggests the importance of the vaccination for the recovery of the ridesourcing services. As for the number of COVID-19 cases, it was found to have a negative impact on the demand reduction levels in Innisfil and a positive effect in Chicago. This is due to the fact that the demand for ridesourcing in Chicago reached its lowest point at the beginning of the pandemic, when both the number of COVID-19 cases and tests were low. Later on when the city conducted more tests and therefore recorded more cases, the demand increased slightly (see Fig 4).

In terms of the relative importance of variables, the results indicated that the stringency index values had the highest impact on the daily reduction levels in Innisfil, followed by the trip characteristics, vaccination rates, and COVID-19-related statistics, respectively. In Chicago, the percent reduction in ridesourcing demand was most influenced by the trip characteristics, followed by the stringency index, COVID-19 variables, and vaccination rates, respectively. The relative importance of the vaccination rates between the two locations further explains the slower recovery rate observed in Chicago for the ridesourcing services. Additionally, this suggests that the recovery of the service could be linked to factors other than vaccination coverage, for example, the economic recovery, the efficacy of the vaccine against the new variants, and individual attitudes and perceptions towards the virus. However, further analysis is needed to understand the factors affecting the recovery of the ridesourcing services. It can also be noticed that the test positivity rate variable was the most important among the COVID-19 related variables in Innisfil, while the daily number of hospitalizations and deaths were the least important. In contrast, the opposite pattern can be observed in Chicago. This can be explained by the low number of people hospitalized or who died from the COVID-19 in Innisfil as compared to Chicago.

## Factors affecting the daily direct demand for ridesourcing services during the pandemic

Here, we explore the impact of the COVID-19 related variables and stringency index, land-use data, socioeconomic and demographic characteristics, as well as the weather data on the direct demand (OD-pair trips) for the ridesourcing services in Innisfil and Chicago during the pandemic.

**Socioeconomic, demographic, and land-use variables.** The results revealed that population density, median income, working-age population, percentage of men, and percentage of married people at the trip origin and destination positively impacted ridesourcing demand in both locations (Fig 6). Besides, smaller households, higher levels of education, and lower percentages of workers commuting by personal vehicles at the trip origin and destination were associated with higher demand. These findings are consistent with those of previous studies for pre-pandemic conditions [58, 59]. However, household size and education factors showed an opposite pattern in Innisfil. Larger households and lower levels of education were associated with higher demand. This variation is mainly due to the subsidized trips offered by the Town Innisfil, which made the service available for a broader segment of the population. Moreover, the OD-pair demand was negatively influenced by the percentage of elderly people at the trip origin and destination in both locations. This finding is consistent with the previous literature indicating that areas with higher percentage of seniors experienced greater mobility reductions during the pandemic [31]. The main reason behind this is that seniors are more likely to get severe symptoms, be hospitalized, or die from the COVID-19 [60].

Furthermore, Innisfil's ridesourcing demand was higher between residential and industrial neighbourhoods, most likely due to work-related trips. In Chicago, the demand for ridesourcing was higher between neighborhoods with governmental and commercial land-use types, probably due to the trips made to and from O'Hare International Airport. Interestingly, the results revealed that large parks in both locations had a high demand for ridesourcing. This is in line with recent findings indicating that the usage of parks and green spaces increased significantly during the pandemic [61]. It is believed that this pattern is related to the fact that during the pandemic, parks were the preferred destination for residents for mental and physical health reasons.

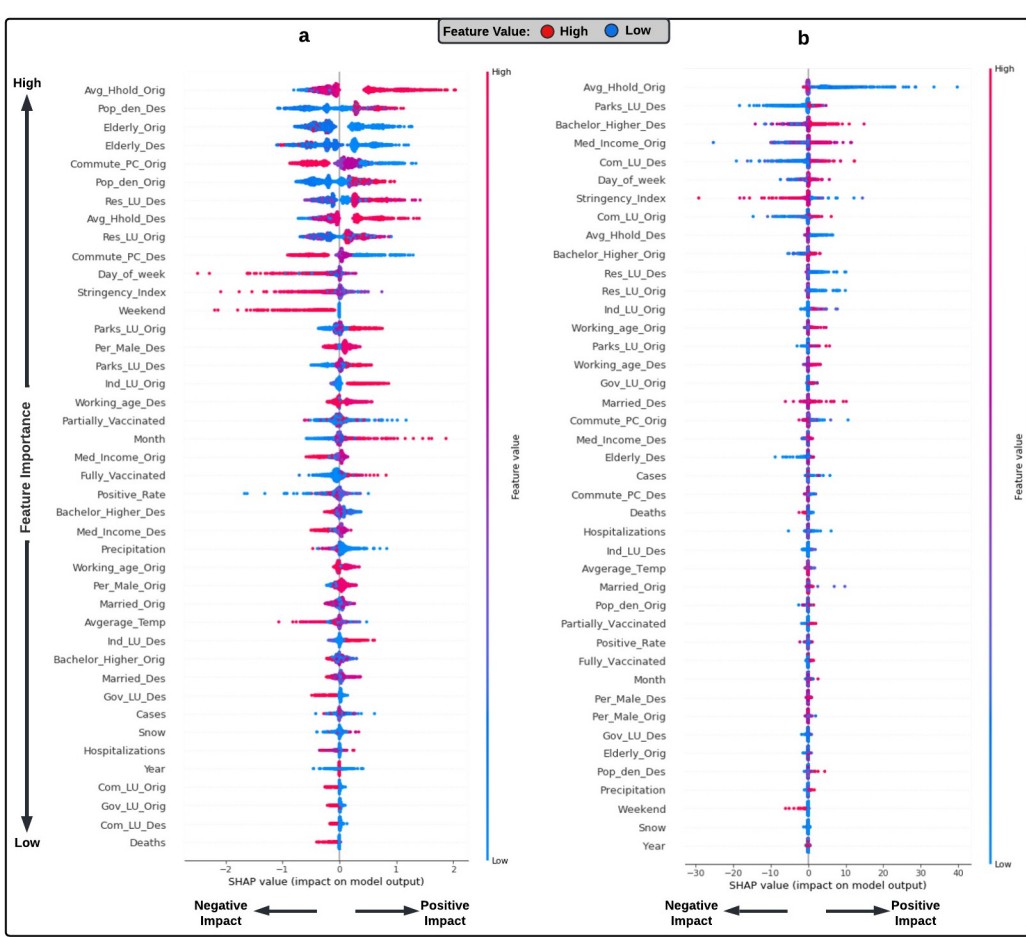

**Fig 6. SHAP summary plot for the direct demand model in (a) Innisfil and (b) Chicago.**

**COVID-19 related variables, vaccination rates, and stringency index.** As expected, the results indicated that the COVID-19 related variables and the stringency index adversely affected the OD-pair trips in both Innisfil and Chicago. The frequency of the ridesourcing trips decreased with the increase in the COVID-19 cases, hospitalization, deaths, test positivity rate, and the stringency index values. In contrast, a positive correlation was found between fully and partially vaccinated individuals and ridesourcing demand. These findings point to the importance of incorporating pandemic-related characteristics into travel behaviour studies.

**Trip and weather variables.** Ridesourcing demand in both locations was negatively associated with the weekend variable and positively associated with the month-of-year variable. Therefore, OD-pair trips were lower on weekends and at the beginning of the year. Taking into account the demand for ridesourcing services was higher on weekends before the pandemic [58], it may be argued that the pandemic has changed the travel behaviour of individuals. Moreover, the impact of the day-of-week variable varied between the two locations, with a negative impact in Innisfil and a positive effect in Chicago. As revealed in this finding, Innisfil's ridesourcing service appears to be used regularly at the beginning of the week for work-related purposes.

On the other hand, the results showed that the ridesourcing demand increased as temperature decreased. This is in line with previous literature and might be the result of individuals switching from non-motorized modes to ridesourcing in cold weather [58]. Although Innisfil and Chicago have similar weather conditions, the demand in Innisfil decreased under rain and increased under snow, whereas the opposite was true in Chicago. The reason for this might be that in rainy and warm weather, individuals in Innisfil are likely to walk or cycle to nearby destinations, while in snowy and icy weather conditions, they switch to ridesourcing services to avoid slipping or being outside in cold. The increase in demand under rain in Chicago might be related to individuals engaging in more activities in warm weather, and the decrease in snowy conditions might be attributed to event cancellations, lower supply, and higher prices.

**Variable importance.**   Overall, the socioeconomic, demographic, and land-use variables had the greatest impact on the OD-pair trips in both locations, followed by the trip characteristics and the stringency index, respectively. These results are consistent with the pre-pandemic conditions, where the demand was most affected by the socioeconomic and demographic characteristics [58]. Innisfil's direct demand was least affected by weather factors and COVID-19 characteristics. Among the socioeconomic and demographic characteristics, the average household size at the trip origin and the population density at the trip destination were the most influential factors. Moreover, the test positivity rate had the highest importance among the COVID-19 related variables. In Chicago, vaccination rates and weather variables had the least influence on direct demand. The average household size at the trip origin as well as the education level at the trip destination were the most important socioeconomic and demographic characteristics. Finally, the daily number of cases was the most influential among the COVID-19 related variables. These findings are in line with our demand reduction model results.

## Discussion and concluding remarks

The on-going COVID-19 pandemic has significantly impacted individual's mobility [14, 31]. Understanding these effects can help planners and policy makers to better anticipate the pandemic's future impacts, which may contribute both to recovery efforts and plan responses to additional waves of COVID-19 or other pandemics. Previous studies, however, have only considered the impacts of COVID-19 on ridesourcing services in early stages and have primarily focused on large cities. This is mainly due to the apparent lack of readily available ridesourcing data. In this study, we examined and compared the impact of the COVID-19 pandemic on ridesourcing services in small towns and large cities using actual data from the Town of Innisfil, Ontario, and the City of Chicago, Illinois. The main factors that governed the selection of these case studies were: a) the availability of ridesourcing data, b) the coverage of both the pre-pandemic and pandemic periods, and c) the location of both case studies within the same region (North America). The two case studies are located in North America, have similar weather conditions, share some sociodemographic characteristics, such as percentage of males, percentage of working age group, and average household size, and have experienced various NPIs during the pandemic. On the flip side, they differ in the land area, population density, the use of ridesourcing services and their operating policies, as well as the severity of NPIs over time.

Overall, the pandemic has affected the usage behaviour of ridesourcing services in both Innisfil and Chicago. The ridesourcing demand was most affected at night and on weekends, as a consequence of closing non-essential businesses. The physical and mental health issues associated with the pandemic resulted in fewer trips in areas with higher proportions of seniors and more trips to parks and green spaces. Moreover, the results revealed that the

demand was adversely affected by the stringency index and COVID-19-related variables, and positively affected by vaccination rates. However, the extent of the effects of pandemic-related factors varied across the two case studies. First, the demand in Innisfil, where ridesourcing services are subsidized and used mainly for work-related purposes, experienced a markedly lower reduction rate, was less affected by the increase in the stringency index, and showed higher recovery rates following each relaxation of restrictions than in Chicago. Second, the low daily number of hospitalizations and deaths (maximum daily values were at 1 and 3, respectively) from COVID-19 in Innisfil, resulted in these factors having the least impact on individuals' use of the service compared with other COVID-19 related variables. In contrast, the positive rate variable, which indicates the level of COVID-19 transmission in the community, had the highest impact on individuals' demand for the service, most likely because it increased their fear of getting infected. The opposite pattern was observed in Chicago, where the COVID-19 hospitalization and death rates were substantially higher than in Innisfil (maximum daily values were at 203 and 58, respectively). The high number of hospitalizations and deaths probably increased individuals' concerns about the consequences of the virus, and, therefore, affected their use of the service most, compared with other COVID-19 variables. Last, compared with Chicago, vaccination rates had a higher impact on the demand for ridesourcing services in Innisfil. It is possible that the low population density in Innisfil, along with the low number of COVID-19 cases, hospitalizations, and deaths, resulted in more people feeling comfortable participating in different activities after receiving the vaccination.

We provide the following recommendations for planners, policymakers, and service providers, which can be useful in recovering ridesourcing services and planning for additional COVID-19 waves or other pandemics:

- Previous literature revealed that public transit services have experienced a significant reduction in demand during the early stages of the pandemic [27, 29]. Governments' policy measures during this period, such as stay at home orders and reduced capacity policies, may have affected the efficiency, operating costs, and performance of public transit services, particularly in low density areas. In light of this, we recommend subsidizing ridesourcing services in the low-density areas rather than operating the existing fixed-route public transit system at high stringency levels. The use of these services requires no capital expenditures and can help limit individuals' interaction. It can also provide essential workers with flexible and convenient transportation, create new jobs for those who lost their jobs due to the pandemic, and support service providers. This suggestion, however, needs further study to verify its feasibility and validity.

- Our results indicated that COVID-19 variables adversely affect individuals' demand for ridesourcing services. Therefore, transportation service providers should ensure that drivers and passengers follow the essential COVID-19 preventative measures to restore individuals' confidence in the safety of ridesourcing services and foster the recovery of the service. Providing users with the opportunity to rate the driver's preventative measures might be useful.

- Our findings revealed that ridesourcing services experienced a repeated pattern of a sharp decrease in demand followed by a gradual recovery. It is possible that this has impacted the number of drivers and their working hours. Thus, transportation service providers should provide a proper incentive strategy for drivers to ensure that there is adequate supply as the demand recovers and that performance remains the same as it was before the pandemic. The demand models obtained in this work might be of use for the service providers in Chicago and Innisfil to anticipate the demand as the policy measures change. Although the models

may not be suitable for ridesourcing services in other locations, they can provide key insights about the factors affecting the demand.

- The use of the pandemic-related factors contributed to interpreting the temporal variations in demand for ridesourcing and understanding their effects. This suggests the importance of incorporating them in future travel behaviour studies.

This research has two main limitations: (a) Innisfil's ridesourcing dataset does not cover the early stages of the pandemic (from March to August 2020), and (b) our findings are based on ridesourcing data from Innisfil and Chicago, therefore, they may not be directly generalized to other towns or cities. Future research may explore the impact of the COVID-19 pandemic on the demand for other modes of transportation as well as the supply of ridesourcing services. Another future direction could be to investigate the factors affecting the recovery of public transit and ridesourcing services.

## Supporting information

**S1 Fig. Pre-pandemic origin-destination (OD) flow patterns for the ridesourcing services in Chicago.** As during the pandemic, high-demand OD pairs were in the downtown area and between the downtown and O'Hare International Airport. The pre-pandemic trip data in Innisfil did not have the required information to develop the OD flow patterns. However, the ridesourcing trip density maps developed by Sweet et al. [62] for the time period from May 2017 to February 2020, showed that the Innisfil's central area had the highest pick-up and drop-off density, which is similar to the patterns observed during the pandemic.
(TIF)

**S1 Table. Optimal hyperparameters set for the models developed in this study.**
(DOCX)

**S2 Table. Model performance.** Across all performance metrics, the models in Chicago had a higher predictive accuracy than those in Innisfil. This is most likely because Chicago's datasets contained more samples.
(DOCX)

## Acknowledgments

We are thankful to the Town of Innisfil and Uber Technologies Inc. for providing us access to the operational data of Innisfil Transit Service that was used in this study. We also thank Nicoleta Hera for her time and effort in helping us improve the quality of this manuscript.

## Author Contributions

**Conceptualization:** Nael Alsaleh, Bilal Farooq.

**Data curation:** Nael Alsaleh.

**Formal analysis:** Nael Alsaleh.

**Investigation:** Nael Alsaleh, Bilal Farooq.

**Methodology:** Nael Alsaleh, Bilal Farooq.

**Software:** Nael Alsaleh.

**Supervision:** Bilal Farooq.

**Visualization:** Nael Alsaleh.

**Writing – original draft:** Nael Alsaleh, Bilal Farooq.

**Writing – review & editing:** Nael Alsaleh, Bilal Farooq.

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
