## [Decision Letter · Decision Letter 0]

31 May 2022

PONE-D-22-08009The impact of COVID-19 pandemic on ridesourcing services differed between small towns and large citiesPLOS ONE

Dear Dr. Alsaleh,

Thank you for submitting your manuscript to PLOS ONE. After careful consideration, we feel that it has merit but does not fully meet PLOS ONE’s publication criteria as it currently stands. Therefore, we invite you to submit a revised version of the manuscript that addresses the points raised during the review process.

Both reviewers recommended major revision. They pointed out insufficient descriptions and interpretations about the results. I agree to them. Please upgrade the manuscript following the comments from the reviewer.

We look forward to receiving your revised manuscript.

Kind regards,

Hironori Kato, Dr. Eng.

Academic Editor

PLOS ONE

3. We note that Figure 3 in your submission contain map images which may be copyrighted. All PLOS content is published under the Creative Commons Attribution License (CC BY 4.0), which means that the manuscript, images, and Supporting Information files will be freely available online, and any third party is permitted to access, download, copy, distribute, and use these materials in any way, even commercially, with proper attribution. For these reasons, we cannot publish previously copyrighted maps or satellite images created using proprietary data, such as Google software (Google Maps, Street View, and Earth). For more information, see our copyright guidelines: http://journals.plos.org/plosone/s/licenses-and-copyright.

 a. You may seek permission from the original copyright holder of Figure 3 to publish the content specifically under the CC BY 4.0 license. 

Reviewers' comments:

Reviewer's Responses to Questions

**Comments to the Author**

1. Is the manuscript technically sound, and do the data support the conclusions?

Reviewer #1: Partly

Reviewer #2: Yes

2. Has the statistical analysis been performed appropriately and rigorously? 

Reviewer #1: I Don't Know

Reviewer #2: Yes

3. Have the authors made all data underlying the findings in their manuscript fully available?

Reviewer #1: Yes

Reviewer #2: Yes

4. Is the manuscript presented in an intelligible fashion and written in standard English?

Reviewer #1: Yes

Reviewer #2: Yes

5. Review Comments to the Author

Reviewer #1: The authors investigated and compared factors influencing the use of RHA in the cities of Chicago, IL and Innisfil, ON during the COVID-19 pandemic using the random forest regression method. The results revealed the uniqueness of RHA trip characteristics in the big and the small cities and showed the different impacts of the COVID-19 pandemic on the use behaviors of RHA in these two cities. Their results can be useful for transport planners and policymakers to better understand how people were dependent on RHA and how the sensitivity of RHA use to the change in travel demand characteristics (e.g., due to the pandemic, policy, new lifestyle). However, several improvements are required before the manuscript can be published.

• What’s the point of comparing big and small cities, and more specifically Innisfil and Chicago which are vastly different, not only in their sizes but also in different countries as well as transportation system infrastructure and travel demand characteristics. It’s no surprise that Covid would affect them in different ways. The analysis of Chicago data alone can potentially provide sufficient scientific contribution. How would data from the tiny Innisfil add to scientific knowledge? Why not the town of the same size in different countries, or different sizes in the same country? You need to provide more convincing justification of the choice of study areas.

• The methodology section needs elaboration. The diagram in Figure 1 should be explained in more detail. Also check for typos. Explanations of specific techniques, including Random forest regression, Bayesian optimization, and technical measurements (e.g., SHAP value, stringency index) are needed. Some equations and detailed descriptions of definition of dependent variables can be helpful.

• Descriptive statistics of data and commentaries are much needed.

• If the term “post-pandemic” was used, the authors should clearly state that which period was considered as post-pandemic.

• Would modeling the reduction of OD-pair trips gain more insights?

• Discussion should be improved. For example, how did the results link to area’s unique characteristics? How the transport policy or plan could be developed based on the findings? Any transportation policy implications with regards to Covid restrictions?

• Figs 2 and 3 are probably switched.

Reviewer #2: This paper aims to investigate the effect of COVID-19 on RS demand. This manuscript is fascinating, and I am impressed with this; however, I have some suggestions that the authors need to act on.

Abstract:

The authors stated that “it is still unclear how the pandemic affected the demand for ridesourcing Services”. In fact, some studies show that the demand for RS decrease due to pandemic (for example, see: https://doi.org/10.1007/s11116-021-10185-5. Due to this, the authors need to revise that sentence.

Introduction

1. “Previous literature revealed that the pandemic has significantly influenced all modes of transportation” � use and frequency of use (please add references)

2. Data collected from November 2018 to August 2021 � Explain a brief of government policies were applied to suppress the COVID-19 during the period in two cities. Clearly, it significantly influences the number of trips. I read that the authors have explained it in lines 122-128. But please explain in more detail because I believe that the government policies during the study period are changing rapidly.

Materials and Methods: Data collection

1. The authors stated that data for pre-pandemic period in Innisfil were obtained from Uber Technologies. Are there any other RS companies instead of Uber in that city? If yes, it could make the biased data. The authors need to explain about this.

2. Line 141-145: make it a sentence

Results

1. The authors stated that they use RF regression algorithms and include four performance metrics to evaluate the predictive accuracy of the models. However, I can not see those performance metrics in the result section. How can the authors ensure the readers and me that their model is correct?

2. The authors need to explain how to understand whether the considered variables significantly influence RH use or not. If I check your discussion, all included variables in Fig. 6 significantly influenced RH use with positive and negative signs. When I use the discrete choice model, for example, each variable has a p-value to determine whether or not a variable is significant.

3. How can the authors state that a variable has the highest effect while other variables have the lowest impact? Does it depend on the values in Fig. 6.? If yes, the authors need to state it.

Discussion and concluding remarks

1. Delete lines 389-394, because you have clearly stated it in the introduction section.

2. More importantly, clearly state how your findings fill the gap of previous findings.

3. The authors mentioned that the study contributes to policymakers as well as service providers to understand the future impact of the pandemic on ridesourcing services in terms of the recovery of the service, preparedness for additional waves of COVID-19 as well as other pandemics. Please add some examples of government policies in the discussion section based on your research findings to strengthen your statements.

6. PLOS authors have the option to publish the peer review history of their article (what does this mean?). If published, this will include your full peer review and any attached files.

Reviewer #1: No

Reviewer #2: No

---

## [Author Response · Author response to Decision Letter 0]

12 Aug 2022

Response to Reviewers' Comments:

Reviewer1:

1.1: Thank you for the comment. We agree that comparing different towns/cities with each other or even including more case studies can provide useful insights. However, the actual data for ridesourcing services is very limited and only provided under license in most cases. Additionally, the motivation for study was the need to understand and compare the impact of COVID-19 across urban areas within the same geographical region. Several factors governed the selection of the case studies, including a) the availability of ridesourcing data, b) the coverage of both the pre-pandemic and pandemic periods, and c) the location of both case studies within the same region. After gaining access to the actual ridesourcing data in the Town of Innisfil, the City of Chicago was selected as a second case study. To the best of our knowledge, the City of Chicago is the only case study, where a publicly available ride sourcing data satisfies the above-mentioned criteria.

The Town of Innisfil and the City of Chicago are located in North America, where the population is concentrated in the largest cities, while rural areas, are sparsely populated. The case studies have similar weather conditions, share some sociodemographic characteristics, such as percentage of males, percentage of working age group, and average household size, and have experienced various NPIs during the pandemic. On the flip side, as you have mentioned, they differ in the land area, population density, the use of ridesourcing services and their operating policies, as well as the severity of NPIs over time. In this study, we examined the impacts of the COVID-19 pandemic in both two case studies and discussed how these differences have resulted in the COVID-19 pandemic having different effects.

The City of Chicago has a rich dataset that, as you stated, provides several insights on how the pandemic and its related characteristics have impacted ridesourcing services. Adding data from Innisfil, however, contributed to a better understanding of the similarities and differences in the demand for ridesourcing services among small towns (low-density area) and big cities (high-density area). As discussed in the results, most of the socioeconomic and sociodemographic characteristics, land use variables, COVID-19 related variables and NPIs, as well as trips characteristics had the same impact on the ridesourcing services in both locations. But the relative importance of these variables differed between the two case studies. Second, it contributed to a better understanding of the pandemic effects on the ridesourcing services, after linking the results to the characteristics of each case study. 

We have added the summary of this discussion at the beginning of the "Data collection" as well as "Discussion and concluding remarks" sections.

1.2: Thank you for the comment, and we apologize for the typos in Figure 1. We have revised the methodology section and made the following changes: a) adding a new subsection "Methods", where we explain the main technical measurements and techniques used in this study, including: Stringency Index, Random Forest, Bayesian Optimization, and SHAP Analysis. b) Using equations to explain the dependent variables of both the percent reduction in daily demand and the direct demand models In "Data analysis and modelling" section. c)Adding more information on how the techniques have been implemented in our study to the last paragraph of "Data analysis and modelling" section. d) Correcting the oversights in Figure 1.

1.3: Thank you for the comment. We have now provided a descriptive statistics for the dependent and explanatory variables in the "Data analysis and modelling" subsection.

1.4: Thank you for the comment. "Post-pandemic" refers to the period following the declaration of COVID-19 as a global pandemic in this study. We have now defined the term in the Introduction Section following its first use.

1.5: Thank you for the comment. We agree that modelling the reduction in the demand at the OD-pair level would provide more insights than modelling at the town/city level. However, the data we would need to develop such models are not available for both case studies, as pre-pandemic data for Innisfil are only available at the aggregate level (town level). Thus, for consistency and accuracy, we modelled the daily reduction in demand at the town and city level. 

1.6: Thank you for the comment. This section has been extensively revised to discuss the results and relate them to the characteristics of each case study, and to provide useful policy recommendations for planners and service providers. 

1.7: Thank you for letting us know. The figures are now arranged correctly.

Reviewer 2:

2.1: Thank you for your comment. We agree that previous studies have examined some aspects of the COVID-19 pandemic's impact on ridesourcing services, but they have not covered all of them. The topic needs to be explored from several perspectives to have a complete understanding of how the pandemic affected the service. These include questions like, did the pandemic effects on ridesourcing services vary temporally and spatially? and why? What factors affect the recovery of the service? What effects has the pandemic had on individual's attitudes and perceptions towards the service? and for how long these effects will last?, and more. Thus, we believe this sentence is still valid, but in order to be more precise, we have modified the first two sentences of the abstract.

2.2: Thank you for the comment. The sentence has been updated and references have been added.

2.3: Thank you for the comment. We have a brief of the policy measures taken by Ontario and Illinois during the study period in the ``Data collection" subsection. 

2.4: Thank you for bringing this to our attention. In fact, Uber Technologies is the only transportation network provider in Innisfil. This is now mentioned in the Data Collection section.

2.5: Thank you for the comment. We now present the time periods in a sentence.

2.6: Thank you for the comment. The optimal hyperparameters as well as the performance metrics of the developed models are provided in the "Supporting information" section. We have referred to these tables in the last paragraph of the "Data analysis and modelling" section. 

We have used several ways to ensure the quality of the modelling results. a) Providing a detailed methodology for data collection, analysis, and modelling. We have also explained the techniques as well as the technical measurements used in the modelling part in the revised version. b) Using a powerful automatic tuning approach (Bayesian Optimization Method) for the hyperparameters of the Random Forest algorithm, which ensures that each model used the hyperparameters that produced the best performance. c) Providing the key performance metrics for each model. d) Using the SHAP analysis technique to interpret the predictions of each model. e) Providing the code used to produce the models presented in this study along with the minimal data.

2.7: Thank you for the comment. SHAP analysis results do not provide insight into whether an explanatory variable impacts the model outcomes significantly, but rather explain its effects and relative importance compared to other explanatory variables. We have now provided more details about the techniques used in the modelling part, including SHAP analysis method, under the "Methods" subsection of the "Materials and methods" section.

2.8: Thank you for the comment. The position of an explanatory variable on the vertical axis indicates its relative importance for the model outcomes compared to other explanatory variables. We provided a description of the SHAP summary plot in the caption of Fig 5, where it first appeared in the manuscript. Moreover, we have now added a legend to both Figs 5 and 6 to make it easier for readers to understand these plots.

2.9: Thank you for the comment. This section has been extensively revised. We have removed lines 389-394, discussed the results and linked them to the characteristics of each case study, explained how our study fill the gap in the literature, and provided useful policy recommendations for planners and service providers.

---

## [Decision Letter · Decision Letter 1]

23 Sep 2022

The impact of COVID-19 pandemic on ridesourcing services differed between small towns and large cities

PONE-D-22-08009R1

Dear Dr. Alsaleh,

We’re pleased to inform you that your manuscript has been judged scientifically suitable for publication and will be formally accepted for publication once it meets all outstanding technical requirements.

Kind regards,

Hironori Kato, Dr. Eng.

Academic Editor

PLOS ONE

Reviewers' comments:

Reviewer's Responses to Questions

**Comments to the Author**

1. If the authors have adequately addressed your comments raised in a previous round of review and you feel that this manuscript is now acceptable for publication, you may indicate that here to bypass the “Comments to the Author” section, enter your conflict of interest statement in the “Confidential to Editor” section, and submit your "Accept" recommendation.

Reviewer #1: All comments have been addressed

Reviewer #2: All comments have been addressed

2. Is the manuscript technically sound, and do the data support the conclusions?

Reviewer #1: Yes

Reviewer #2: Yes

3. Has the statistical analysis been performed appropriately and rigorously? 

Reviewer #1: Yes

Reviewer #2: Yes

4. Have the authors made all data underlying the findings in their manuscript fully available?

Reviewer #1: No

Reviewer #2: Yes

5. Is the manuscript presented in an intelligible fashion and written in standard English?

Reviewer #1: Yes

Reviewer #2: Yes

6. Review Comments to the Author

Reviewer #1: All comments have been satisfactorily addressed. There are some duplicated words and typos. Please check carefully.

Reviewer #2: The study has well addressed the comments raised in the previous round and has substantially improved a lot. I am confident that this study will be meaningful for both research and practise. I recommend to the editor that this paper can be accepted for publication.

7. PLOS authors have the option to publish the peer review history of their article (what does this mean?). If published, this will include your full peer review and any attached files.

Reviewer #1: No

Reviewer #2: No

---

## [Editor Report · Acceptance letter]

6 Oct 2022

PONE-D-22-08009R1 

The impact of COVID-19 pandemic on ridesourcing services differed between small towns and large cities 

Dear Dr. Alsaleh:

I'm pleased to inform you that your manuscript has been deemed suitable for publication in PLOS ONE. Congratulations! Your manuscript is now with our production department. 

Kind regards, 

on behalf of

Dr. Hironori Kato 

Academic Editor

PLOS ONE